# Optimal Engagement-Diversity Tradeoffs
# in Social Media

## ABSTRACT

Social media platforms are known to optimize user engagement with the help of algorithms. It is widely understood that this practice gives rise to echo chambers — users are mainly exposed to opinions that are similar to their own. In this paper, we ask whether echo chambers are an inevitable result of high engagement; we address this question in a novel model. Our main theoretical results establish bounds on the maximum engagement achievable under a diversity constraint, for suitable measures of engagement and diversity; we can therefore quantify the worst-case tradeoff between these two objectives. Our empirical results, based on real data from Twitter, chart the Pareto frontier of the engagement-diversity tradeoff.

**ACM Reference Format:**

Anonymous Author(s). 2023. Optimal Engagement-Diversity Tradeoffs in Social Media. In *Proceedings of ACM Conference (Conference'17)*. ACM, New York, NY, USA, 13 pages. https://doi.org/10.1145/nnnnnnn.nnnnnnn

## 1 INTRODUCTION

It is no secret that social media companies heavily rely on algorithms to optimize user engagement. This practice has a well-documented dark side that is widely scrutinized and debated. For example, writing recently in the New York Times, the technology pioneer Jarron Lanier coins the term "Twitter poisoning" to describe "a side effect that appears when people are acting under an algorithmic system that is designed to engage them to the max" [19].

Perhaps the main reason that optimized engagement is so broadly decried is that it may lead to increased political polarization through the formation of echo chambers, where users are only exposed to viewpoints and opinions that closely align with their own. In the pages of the Washington Post, [16] lament that "the features that facilitate a right-wing echo chamber on Facebook — such as [...] how the algorithms work to maximize engagement — are intentional choices."[1] This statement is supported by a paper by the same authors [17] and grounded in a large, important body of prior work [7, 21, 24, 25].

In several studies, however, there is somewhat mixed evidence for the relation between algorithms, the diversity of content users consume, and political polarization [5, 9]. In fact, an influential

[1]Ironically, they give Twitter as a positive example, at least in terms of the stated intentions of its founder and former CEO, Jack Dorsey.

paper casts doubt on the very idea that eliminating echo chambers and showing more diverse information sources leads to reduced polarization [4]. But even the lead author of that paper concedes that, on Twitter, "breaking up the echo chambers that prevent cross-party discussion about market-based solutions to climate change, for example, might be more successful" than having broad conversations about politics [3].

This debate notwithstanding, it seems that academics and pundits largely agree on one underlying assumption: There is a tradeoff between user engagement and the diversity of information they are exposed to. In other words, if a social media platform wishes to maximize engagement and optimize its revenue, it would necessarily have to expose users to the posts or tweets they are most likely to engage with, thereby limiting diversity of information and creating echo chambers.

In this paper, we aim to *quantify* the engagement-diversity tradeoff. Our high-level research question is this: *How much engagement must be sacrificed in order to guarantee a given level of diversity of information?* We are particularly interested in identifying scenarios where diversity of information comes at little cost to engagement, as in such scenarios, it is more likely that social media platforms would be willing to break up echo chambers.

*Our approach and results.* We base our terminology on Twitter,[2] but our model and analytical results are relevant to most social media platforms, including Facebook. As usual, we represent the social network as a directed graph,[3] where the nodes are users and an edge from $i$ to $j$ means that $i$ follows $j$. We also assume that tweets are partitioned into $T$ types, and each user $i$ has probability $p_{ti}$ of retweeting a tweet of type $t$.

We model the propagation of tweets in the network as a discrete-time (Markov decision) process where, in each step, users are exposed to tweets from their followees, as well as tweets directly shown to them by the platform. This latter component is the algorithmic *injection policy*, which presents to each user tweets of different types, subject to a budget constraint. Note that this modeling choice matches Twitter's default view, where users are shown a combination of tweets shared by their followees and some algorithmically selected tweets that originate in non-followees. The injection policy itself is time-independent, that is, users are exposed to the same mixture of types in each round; as we prove, this is without loss of generality (assuming that the network and retweet probabilities are fixed, of course).

We can now quantify *engagement* by measuring the number of retweets in the system (in the limit as the number of rounds grows). This is a natural measure in our simple model; a more elaborate model may take into account the affinity of users to different tweet types; as we discuss in Section 5, our results extend

[2]While Twitter has recently been rebranded as X, we prefer to keep the old terminology (e.g., "tweets") for historical consistency.
[3]For Facebook we would simply have bidirectional edges between friends.

to this setting. There are many ways to quantify *diversity* in our model; our measure is the minimum, across users $i$ and tweet types $t$, of the number of tweets of type $t$ seen by user $i$ (in the limit as the number of rounds grows). This is a rather onerous choice, as it requires that every user be exposed to every tweet type; such a pessimistic view means that any positive results are especially robust.

To analyze the tradeoff between engagement and diversity, we are interested in two injection policies: the one that maximizes engagement and the one that maximizes engagement subject to achieving at least $\delta$-diversity for a given $\delta \geq 0$. The *cost of $\delta$-diversity*, then, is the fraction of the engagement of the former policy that is sacrificed by employing the latter policy.

Our main theoretical result is an upper bound on the cost of $\delta$-diversity. Assuming the average retweet probability of each user is at least $\alpha$ and each retweet probability is at most $\beta > 0$, and that $\delta \leq 1/T$, the cost of $\delta$-diversity is at most $T\delta(1 - \alpha/\beta)$, and this bound is tight. Qualitatively, the implication is that with a user base that is generally engaged (high average retweet probability $\alpha$ compared to the maximum $\beta$), the cost of $\delta$-diversity is small. This bound is both trivial and tight in the special case where the graph is empty; our result is encouraging in that it demonstrates that the very same bound still holds despite the nontrivial complications arising from the dynamics of retweets in a general social network.

To obtain a more nuanced understanding of the engagement-diversity tradeoff in practice, we also conduct experiments on a large dataset from Twitter. We process the data to extract the social network graph and, based on hashtags, infer four types of tweets and their associated retweet probabilities. We then measure the cost of $\delta$-diversity as the retweet probabilities are scaled up. The results show that the practical tradeoff is far better than the worst-case bound, and that the cost of diversity is typically (though, surprisingly, not always) monotonically decreasing in the magnitude of retweet probabilities. Finally, we discuss how a policy-maker can operationalize these results.

*Related work.* Needless to say, the literature on social networks and recommender systems is vast, and there is a large body of work on diversity in recommender systems, specifically; see, e.g., the survey by Kunaver and Požrl [18].

Perhaps the most relevant papers are those that consider the impact of algorithms on diversity through field experiments [2, 13, 20]. In particular, Holtz et al. [13] study the engagement-diversity tradeoff via a field experiment on Spotify. Their control and treatment groups were given podcast recommendations to maximize engagement; in the case of the treatment group, the recommendation algorithm was personalized, whereas, in the case of the control group, recommendations were based on demographics. Treatment significantly increased engagement and significantly decreased diversity (measured through the category tags of podcasts). The authors conclude that "these findings highlight the need for academics and practitioners to continue investing in personalization methods that explicitly take into account the diversity of content recommended."

A bit further afield, Huszár et al. [15] report results from a field experiment on Twitter, where the control group was shown tweets in reverse chronological order, without algorithmic personalization.

They find evidence for algorithmic amplification of certain political groups; specifically, they conclude that the mainstream political right enjoys higher algorithmic amplification than the mainstream political left. This paper reinforces the connection between algorithms and political polarization on Twitter, but it does not examine the engagement-diversity tradeoff.

It is also worth mentioning that our model is inspired by linear models of social learning such as that of DeGroot [8]. But there are fundamental differences. On a conceptual level, models of social learning typically focus on the convergence of opinion dynamics and the impact of the underlying graph structure. There is typically no decision making during the process. By contrast, in our model we are dealing with an *optimization* problem faced by a central planner. Moreover, we are interested in the tradeoff between novel measures of engagement and diversity. On a technical level, dynamics in a model like DeGroot are captured by a Markov *chain*, whereas our model gives rise to a Markov *decision process*, due to the injection policy.

## 2 MODEL AND MACHINERY

In this section, we introduce our stylized model of the Twitter social network, its dynamics, and key definitions. Additionally, we describe a computational framework for analyzing the engagement-diversity tradeoff and provide theoretical results in support of the robustness of our modeling choices.

*Social network instance.* There is a set of $n$ *users* denoted $[n] = \{1, \ldots, n\}$. As is standard in social networks, users may follow each other. We represent this in the canonical way using a *follower graph* $G = ([n], E)$ with users as vertices and a (directed) edge $(i, j) \in E$ present when user $i$ follows user $j$. We use FOLLOWING$(i)$ to denote the number of users $i$ follows (i.e., $i$'s outdegree in $G$).

There are $T$ *types* of tweets $[T] = \{1, \ldots, T\}$ (indexed by $t$) users can view in their feeds of content. Users' feeds are determined by what type of tweets the users they follow engage with. By retweeting, the user propagates this type through the follower graph. In the next timestep, followers of the user will be able to view and further distribute the tweets. Thus, specific types of content can spread through the network from a small set of initial adopters to a potentially much larger group.

In more detail, it is assumed that upon seeing a tweet, each user has a probability of retweeting this tweet determined by its type. This is denoted by a type's *retweet probability vector* $\mathbf{p}_t \in [0, 1)^n$ where $p_{ti}$ is the probability user $i$ retweets a tweet of type $t$. The collection of retweet probability vectors of all types is denoted by $\mathbf{p} = (\mathbf{p}_t)_{t \in [T]}$. We assume that followers do not get the exact same tweet as the original tweet, in order to avoid addressing the case of a retweet loop in this theoretical framework. Instead, retweeting propagates the type and not its specific realization.

*States.* A *type state* $\mathbf{x}_t^{(k)} \in \mathbb{R}^n$ represents the expected number of tweets of type $t$ being seen by the users. The component $x_{ti}^{(k)}$ is the (expected) number of tweets of type $t$ seen by a user $i$ at time $k$. We use $\mathbf{x}^{(k)} = (\mathbf{x}_t^{(k)})_{t \in [T]}$ to denote the collection of all type states and simply call it a *state*. We will use $\mathbf{x}$ to refer to an unparameterized state (without a timestep).

*User feed.* In our model, there are two ways a tweet may end up in a user's state (i.e., be seen on their feed), either directly from another user they follow or *injected* by the social network. In general, we will assume that the former is exogenous and given as part of the system while the latter is a policy we (as the social network) have control over. To understand the former, suppose a user $i$ follows a user $j$ and $j$ sees a tweet of type $t$ at time $k$. The probability that $i$ sees this tweet at time $k + 1$ we assume to be $\frac{p_{tj}}{\text{FOLLOWING}(i)}$, that is, it is the probability that $j$ retweets this tweet scaled down by the number of users $i$ follows. Observe that this is well-defined as $\text{FOLLOWING}(i) \geq 1$ by the assumption that $i$ follows $j$. The scaling down is to account for the fact that if $i$ follows many users, they will not necessarily see all the retweets in their feed.[4] We use a *type matrix* $\mathbf{A}_t \in \mathbb{R}^{n \times n}$ to store these seen probabilities, where

$$A_{tij} = \begin{cases} \frac{p_{tj}}{\text{FOLLOWING}(i)} & \text{if } i \text{ follows } j \\ 0 & \text{otherwise} \end{cases}.$$

We again use $\mathbf{A} = (\mathbf{A}_t)_{t \in [T]}$ to denote the collection of all type matrices. Note that if the state for type $t$ at time $k$ is $\mathbf{x}_t^{(k)}$, then at time $k + 1$, each user will see $\mathbf{A}_t \mathbf{x}_t^{(k)}$ based on retweets only.

To represent a social network's injection, we define an injection policy $\mathbf{b} = (\mathbf{b}_t)_{t \in [T]}$ where each $\mathbf{b}_t \in [0,1]^n$. The component $b_{ti}$ represents the expected number tweets shown to user $i$ of type $t$. In addition, we require that for each user $i$, $\sum_t b_{ti} \leq 1$, i.e., only at most one tweet "unit" can be injected at each time step.

*Dynamics and limiting behavior.* For an injection policy $\mathbf{b}$, we obtain the following dynamics on tweets seen in the system. At time 0 for each type $t$, we simply have $\mathbf{x}_t^{(0)} = \mathbf{b}_t$. For all times $k \geq 0$, we have $\mathbf{x}_t^{(k+1)} = \mathbf{A}_t \mathbf{x}_t^{(k)} + \mathbf{b}_t$; in words, the tweets seen by users in time $k + 1$ are retweets by others they follow along with direct injections to them. Abusing notation slightly, we allow matrix and vector operations to work over collections, i.e., writing $\mathbf{x}^{(k+1)} = \mathbf{A}\mathbf{x}^{(k)} + \mathbf{b}$ to refer to all types. Notice that for each type individually, this is a standard linear dynamical system. However, this formulation is unusual because the constraint on policies $\mathbf{b}$ (one unit per user) is across types, creating interdependences.

Unraveling the recursion, we see that the $k$'th timestep can be written as $\mathbf{x}^{(k)} = \sum_{\ell=0}^{k-1} \mathbf{A}^\ell \mathbf{b}$ and, by linearity, it can be written as $\mathbf{x}^{(k)} = (\sum_{\ell=0}^{k-1} \mathbf{A}^\ell)\mathbf{b}$. Since the sum of each row of $\mathbf{A}_t$ is strictly less than one, the spectral radius of each type, $\rho(\mathbf{A}_t)$, is less than one (this follows from, e.g., the Gershgorin circle theorem). This implies that the limit $\lim_{k \to \infty} \sum_{\ell=0}^k \mathbf{A}^\ell$ exists and approaches $(\mathbf{I} - \mathbf{A})^{-1}$, where $\mathbf{I}$ is the identity matrix [14]. Since this limit matrix will come up so often, we will use the notation $\mathbf{A}_t^* = (\mathbf{I} - \mathbf{A}_t)^{-1}$ and similarly $\mathbf{A}^* = (\mathbf{I} - \mathbf{A})^{-1}$. However, using this detail, we see that the state also converges to a "limiting state" $\mathbf{A}^*\mathbf{b}$. We will use $\mathbf{x}(\mathbf{b}) = \mathbf{A}^*\mathbf{b}$ to denote the limiting state of policy $\mathbf{b}$ (recall this really means the collection of $\mathbf{x}(\mathbf{b})_t = (\mathbf{A}_t^*)^{-1}\mathbf{b}_t$).

---

[4]This scaling down is admittedly a controversial modeling choice. One could alternatively assume that each user has a fixed "attention budget," but that would lead to nonlinear dynamics. Scaling down by FOLLOWING($i$) is a way of realistically bounding the number of tweets users see — after all, if a user follows thousands of people, they will not have time to peruse all their tweets — while preserving the linearity of the model. It thus strikes a good balance between realism and technical tractability.

*Engagement and diversity.* Two desirable properties guide our analysis. The *engagement* of a state $\mathbf{x}$ is denoted $\text{ENG}(\mathbf{x}) = \sum_t \langle \mathbf{p}_t, \mathbf{x}_t \rangle$. It captures the expected number of retweets generated by the state $\mathbf{x}$. In some sense, this assumes that a user's "engagement" with a tweet is simply the likelihood of them retweeting it. However, as we discuss in Section 5, the coupling of engagement and retweet probability is unnecessary; we primarily do so for ease of presentation as it seems like a reasonable choice for such an engagement parameter. The *diversity* of a state $\mathbf{x}$ is denoted $\text{DIV}(\mathbf{x}) = \min_{t \in [T], i \in [n]} x_{ti}$, i.e., the fewest (expected) tweets of any type seen by any user. We say that a state *satisfies $\delta$-diversity* if $\text{DIV}(\mathbf{x}) \geq \delta$. Additionally, we extend the notions of engagement and diversity to injection polices by simply having them operate on their limiting state. Formally, we have $\text{ENG}(\mathbf{b}) = \text{ENG}(\mathbf{x}(\mathbf{b}))$, $\text{DIV}(\mathbf{b}) = \text{DIV}(\mathbf{x}(\mathbf{b}))$, and $\mathbf{b}$ satisfies $\delta$-diversity exactly when $\mathbf{x}(\mathbf{b})$ does.

We let $OPT^{eng}(G, \mathbf{p})$ be the optimal engagement for graph $G$ and retweet probabilities $\mathbf{p}$, that is, the maximum over injection policies $\mathbf{b}$ of $\text{ENG}(\mathbf{b})$. We sometimes will write $OPT^{eng}$ if $G$ and $\mathbf{p}$ are clear from the context.

To understand the engagement-diversity tradeoff, we are especially interested in the optimal engagement achievable under a diversity constraint. We denote this by $OPT^\delta(G, \mathbf{p})$, parameterized by $\delta$, that is, the maximum over injection polices $\mathbf{b}$ with $\text{DIV}(\mathbf{b}) \geq \delta$ of $\text{ENG}(\mathbf{b})$. We will again sometimes write $OPT^\delta$ if $G$ and $\mathbf{p}$ are clear from context. Notice that it is always feasible to guarantee $\delta$-diversity for $\delta \leq 1/T$ since the policy with $b_{ti} = 1/T$ for all $i$ and $t$ achieves this. However, for $\delta > 1/T$, there are instances where no policy achieving $\delta$-diversity exists. Hence, from now on, we will only focus on $\delta \leq 1/T$.

*Cost of $\delta$-diversity.* Finally, we define $\text{COST}^\delta(G, \mathbf{p}) = 1 - \frac{OPT^\delta}{OPT^{eng}}$. This captures the multiplicative loss on optimal engagement by imposing $\delta$-diversity, i.e., a cost of .2 for $\delta = .1$ means that 20% of engagement is lost by enforcing .1-diversity. From another perspective, $1 - \text{COST}^\delta(G, \mathbf{p})$ as a function of $\delta$ plots the Pareto frontier of the trade-off between engagement and a given diversity level.

*Optimizing engagement and diversity.* As it turns out, computing $OPT^{eng}$ amounts to solving a linear program. Namely, we have that $\text{ENG}(\mathbf{b}) = \sum_t \langle \mathbf{p}_t, \mathbf{x}(\mathbf{b})_t \rangle = \sum_t \langle \mathbf{p}_t, \mathbf{A}_t^*\mathbf{b}_t \rangle = \sum_t p_t^\top \mathbf{A}_t^*\mathbf{b}_t$. This is a linear objective in variables $b_{ti}$ for $t \in [T]$ and $i \in [n]$. This objective will come up quite often throughout our analysis, so we introduce the notation $\mathbf{c}_t = (p_t^\top \mathbf{A}_t^*)^\top$ to be the vector of coefficients on the $\mathbf{b}_t = (b_{t1}, \ldots, b_{tn})$ variables. As before, we use the notation $\mathbf{c} = (\mathbf{c}_t)_{t \in [T]}$. We can interpret a value $c_{ti}$ as the total engagement generated in the system by injecting a unit of type $t$ to user $i$. This allows us to write the engagement as $\text{ENG}(\mathbf{b}) = \sum_t \mathbf{c}_t^\top \mathbf{b}_t$.

Since the constraints of being a valid injection policy are also linear, we can write the whole program as

$$\text{maximize:} \quad \sum_t \mathbf{c}_t^\top \mathbf{b}_t$$

$$\text{subject to:} \quad \sum_t b_{ti} \leq 1, \quad i \in [n]$$

$$b_{ti} \geq 0 \qquad t \in [T], i \in [n].$$

We will refer to this linear program as the *engagement-optimal program*. An interesting observation is that the optimal value and

solutions of the program have a simple closed form. Notice that there are no constraints involving distinct users; the only constraint is that for each user, the total injection is at most one. Hence, the optimal policy is to spend this budget of one only on the tweet type $t$ with the largest objective coefficient $c_{ti}$. In other words, an optimal policy is to: (1) for each user $i$, set a single $b_{ti}$ for a type $t$ maximizing $c_{ti}$ to 1 (or any linear combination of maximizing types), and (2) set all other $b_{t'i}$ to 0. This achieves engagement $\sum_i \max_t c_{ti}$.

Things become less straightforward if we wish to optimize engagement subject to $\delta$-diversity, that is, if we wish to compute $OPT^\delta$. The $\delta$-diversity constraint is also linear, so this remains a linear program, identical to the previous, with the added constraint $(\mathbf{A}_t^* \mathbf{b}_t)_i \geq \delta, t \in [T], i \in [n]$. We refer to this program as the $\delta$-*diversity program*. Unlike before, however, it does not seem to have a concise closed form.

*Robustness of the modeling choices.* One modeling choice that may initially seem unnatural is to define engagement and diversity in the limiting state. An alternate formulation is to have a time horizon $K$ and consider what occurs at each timestep. For example, one could define engagement to be the average (or equivalently sum) engagement over all timesteps, i.e., $\frac{1}{K+1} \sum_{k=0}^{K} \text{ENG}(\mathbf{x}^{(k)})$. Similarly, one could instead require that $\delta$-diversity be satisfied at *every* timestep rather than just in the limit. In such a model, requiring injection policies to be identical in every timestep may be overly restrictive. A priori, it seems plausible that substantially better policies exist that change over time; for example, they can oscillate between different injections or modify what they inject once certain levels of diversity have spread through the network. Hence, we could even allow injection policies to be *time-dependent*, changing what they inject depending on the timestep (in contrast to our *time-independent* program solutions). However, as we formalize and prove in Appendix A, the time-independent engagement policies computed by our programs remain nearly optimal with these alternative definitions; this holds even when compared to the more powerful time-dependent strategies. We therefore expect such a model to lead to qualitatively similar results.

*Relation to the Twitter recommendation algorithm.* In March 2023, Twitter published its recommendation algorithm.[5] Twitter shows tweets to a given user according to two criteria: (i) tweets from users they follow, and (ii) tweets that are selected by the aforementioned recommendation algorithm in order to maximize the user's engagement. Currently, Twitter has a separate feed for each of these criteria (the "following" feed and the "for you" feed), while we consider only a single feed that is generated using both criteria. This simplifies exposition and is conceptually similar, assuming Twitter users look at both feeds. In our dynamics, $\mathbf{x}_t^{(k+1)} = \mathbf{A}_t \mathbf{x}_t^{(k)} + \mathbf{b}_t$, (i) corresponds to the first term and (ii) corresponds to the second term.

The Twitter recommendation algorithm extracts a large number of features for each tweet and user in order to predict the engagement a tweet will generate for a given user; it then selects the

---
[5]https://blog.twitter.com/engineering/en_us/topics/open-source/2023/twitter-recommendation-algorithm

tweets that maximize it. Among other features, the Twitter algorithm extracts communities from the social network (the largest communities are "pop," "news," and "soccer") and then represents users and tweets in terms of their affinities with these communities. There is a natural correspondence to our simplified model, with the features of a tweet summarized as its type $t \in [T]$. Interestingly, then, the Twitter recommendation algorithm selects the injected tweets $\mathbf{b}$ by maximizing $\text{ENG}(\mathbf{b}) = \sum_t \mathbf{c}_t^\top \mathbf{b}_t$, which is a myopic optimization of engagement, i.e., it does not take into account future engagement produced by an injection, nor does it ensure diversity.

## 3 THEORETICAL BOUNDS ON THE ENGAGEMENT-DIVERSITY TRADEOFF

We now turn to providing bounds on the cost of $\delta$-diversity. In order to prove upper bounds, rather than focusing on optimal injection policies, we consider algorithms that, while not optimal, are easier to analyze. We begin this section by defining two. To do so, recall that $c_{ti}$, the coefficient in the optimal programs, represents engagement generated in the limiting state by injecting a unit of type $t$ tweet to user $i$. For each user $i$, let $f_i \in \arg\max_t c_{ti}$ be a tweet type generating maximal engagement. Additionally, recall that $OPT^{eng} = \sum_i \max_t c_{t,i}$, achieved by injecting a unit of $f_i$ to each user $i$.

**Definition 1.** The $\delta$-*uniform policy* for each user $i$ injects $\delta$ of each type $t$ and spends the remaining $1 - T\delta$ budget on $f_i$. More formally, $b_{ti} = \delta$ for $t \neq f_i$ and $b_{f_i i} = 1 - (T-1)\delta$.

This policy is $\delta$-diverse as it directly injects at least $\delta$ of every type to all users. Using the $\delta$-uniform policy, we can immediately derive a worst-case bound on $\text{COST}^\delta(G, \mathbf{p})$ for all graphs $G$ and retweet probabilities $\mathbf{p}$. Indeed, regardless of the underlying values, by injecting $1 - (T-1)\delta$ units of $f_i$, we have that it achieves engagement at least $(1 - (T-1)\delta) \sum_i \max_t c_{ti} = (1 - (T-1)\delta)OPT^{eng}$. Since $OPT^\delta$ must be at least the engagement of this policy, we have

$$\text{COST}^\delta(G, \mathbf{p}) = 1 - \frac{OPT^\delta}{OPT^{eng}} \leq (T-1)\delta \qquad (1)$$

as a worst-case bound.

Note that the bound of Equation (1) is, in some cases, tight. Indeed, consider an empty graph where all users have a positive retweet probability for only one type. In an empty graph, the limiting state is exactly equal to the injection policy. Hence, to achieve $\delta$-diversity, it is necessary to inject $\delta$ of all types to everybody, but this means only a $(1 - (T-1)\delta)$ fraction of the policy can be spent on types from which users derive any engagement.

However, to get beyond this worst-case bound, we need to analyze slightly more intricate policies. The next policy is based on the following idea: Suppose we wish to inject tweets such that, in the limiting state, every user sees *exactly* $\delta$ of each type. Notice that computing this policy is not, in fact, too difficult. For a fixed type $t$, the injected $\mathbf{b}_t^{ex}$ would need to satisfy $\mathbf{A}_t^* \mathbf{b}_t^{ex} = \delta \mathbf{1}$. Expanding the definition $\mathbf{A}_t^* = (\mathbf{I} - \mathbf{A}_t)^{-1}$ and multiplying on both sides, we have that $\mathbf{b}_t^{ex} = \delta(\mathbf{1} - \mathbf{A}_t \mathbf{1})$. Hence, we have that $b_{ti}^{ex} = \delta\left(1 - \sum_j A_{tij}\right)$. It is not immediately obvious that the collection $\mathbf{b}^{ex}$ is a valid injection policy; however, we can prove this is the case. Indeed, notice

each $b_{ti}^{ex} \le \delta$ because entries in $\mathbf{A}$ are nonnegative, so by our restriction that $\delta \le 1/T$, $\sum_i b_{ti}^{ex} \le 1$. Further, our normalization by FOLLOWING($i$) in $\mathbf{A}_t$ ensures that $\sum_j \mathbf{A}_{tij} \le \max p_{ti} \le 1$, so each $b_{ti}^{ex} \ge 0$.

Since this sum $\sum_j \mathbf{A}_{tij}$ will come up frequently, we define $\text{INC}_{ti} = \sum_j \mathbf{A}_{tij}$. Intuitively, $\text{INC}_{ti}$ is the "incoming weight" from all the users $i$ follows. If all users were shown exactly 1 unit of type $t$ at time 0, then, in the next time step, $i$ would see $\text{INC}_{ti}$ units. It follows that, in a steady state where all users see $\delta$ of type $t$, to ensure they see $\delta$ of type $t$ at the next time step, they must be injected $\delta(1 - \text{INC}_{ti})$.

We can now use $\mathbf{b}^{ex}$ to define a new injection policy.

**Definition 2.** The $\delta$-exact policy first injects $b^{ex} = \delta(1 - \text{INC}_{ti})$ of type $t$ to each user $i$ and then spends the remaining $1 - \delta \sum_t (1 - \text{INC}_{ti})$ on $f_i$. More formally, $b_{ti} = \delta(1 - \text{INC}_{ti})$ for $t \ne f_i$ and $b_{f_i i} = 1 - \delta(T - 1 - \sum_{t \ne f_i} \text{INC}_{ti})$.

Since the $\delta$-exact policy is injecting only more than $\mathbf{b}^{ex}$, clearly all users see at least $\delta$ of each type in the limit, so it must be $\delta$-diverse. Using the $\delta$-exact policy, we derive what we view as our main theoretical result.

THEOREM 1. *Fix constants $\alpha \le \beta$ with $\beta > 0$. For all graphs $G$ and retweet probabilities $\mathbf{p}$ such that for each user $i$ (1) their average retweet probability is at least $\alpha$, i.e., $\frac{1}{T} \sum_t p_{ti} \ge \alpha$, and (2) their maximum retweet probability is at most $\beta$, i.e., $\max_t p_{ti} \le \beta$, it holds that $\text{COST}^\delta(G, \mathbf{p}) \le \min \left\{ T\delta \left(1 - \frac{\alpha}{\beta}\right), (T-1)\delta \right\}$. Further, for all $\alpha \le \beta$ with $\beta > 0$, there are instances $G$ and $\mathbf{p}$ satisfying (1) and (2) such that the inequality is tight.*

PROOF. Fix some $\alpha$ and $\beta$, a graph $G$, and retweet probabilities $\mathbf{p}$ satisfying the theorem requirements. Notice that the upper bound of $(T-1)\delta$ follows from the theoretical worst-case discussed above. So for this proof, we show an upper bound of $T\delta(1 - \frac{\alpha}{\beta})$. To prove the theorem, we analyze the engagement derived from the $\delta$-exact policy. We partition the users $V = I \sqcup O$ where $I$ is the set of *inside* users that follow at least one other and $O$ is the set of *outside* users that do not follow anybody. For each outside user $i \in O$, since they do not follow anybody, the policy spends the entire $\delta$ on each type in the first part, and then spends the remaining $1 - T\delta$ on $f_i$. For the inside users $i \in I$, as the average retweet probability for each user they follow is at least $\alpha$, the sum of all incoming edges (even after scaling down by the number of followers) is at least $T\alpha$. Hence, there is a remaining budget of at least $1 - T\delta + T\delta\alpha$ to spend on $f_i$.

Next, let us consider the engagement derived. Recall that $OPT^{eng} = \sum_{i \in [n]} \max_t c_{ti}$. We define $E_I = \sum_{i \in I} \max_t c_{ti}$ and $E_O = \sum_{i \in O} \max_t c_{ti}$ to be engagements derived from injecting to inside and outside users respectively. Note that $OPT^{eng} = E_I + E_O$. Now let us consider the engagement of the $\delta$-exact policy. Notice that by the first part alone, each user sees $\delta$ of each type, and by the $\alpha$ lower bound, they must derive $\delta T\alpha$ engagement from this. In the second part, outside users $i$ contribute $(1 - T\delta) \max_t c_{ti}$ and inside users $i$ contribute at least $(1 - T\delta + T\delta\alpha) \max_t c_{ti}$. Putting this together, we have that the total engagement is at least

$$\underbrace{nT\delta\alpha}_{\text{First part}} + \underbrace{(1 - T\delta)E_O}_{\text{Outside user second part}} + \underbrace{(1 - T\delta + T\delta\alpha)E_I}_{\text{Inside user second part}} \quad (2)$$

Our goal then is to show that (2) is at least

$$\left(1 - T\delta \left(1 - \frac{\alpha}{\beta}\right)\right) OPT^{eng}.$$

We begin by showing that

$$(1 - \beta)E_I + E_O \le \beta n. \quad (3)$$

To that end, we consider a modified instance $(G, \mathbf{p}')$ where the graph remains the same, but we set $\mathbf{p}'$ such that $p'_{ti} = \beta$ for all $t$ and $i$. Notice that this has only increased the values of $\mathbf{p}$. Let $\mathbf{c}'$ be the corresponding $\mathbf{c}$ values and define $E'_I$ and $E'_O$ analogously using $\mathbf{c}'$. Since increasing retweet probabilities can only increase the values $c_{ti}$, this also holds for $E_I$ and $E_O$, so $(1 - \beta)E_I + E_O \le (1 - \beta)E'_I + E'_O$. Hence, it is sufficient to show that $(1 - \beta)E'_I + E'_O \le \beta n$.

In this modified instance, all types are symmetric, so $c'_{t_1 i} = c'_{t_2 i}$ for all types $t_1$ and $t_2$. Hence, the vector $(\max_t c_{t1}, \ldots, \max_t c_{tn}) = \mathbf{c}'_1$, i.e., the vector for type 1 (or any $\mathbf{c}_t$ vector for that matter). Let $\mathbf{A}'_1$ be the type matrix of type 1 in the modified instance, so $A'_{1ij} = \frac{\beta}{\text{FOLLOWING}(i)}$ if $i$ follows $j$ and 0 otherwise. Recall that $(\mathbf{c}'_1)^\top = (\beta\mathbf{1})^\top (\mathbf{I} - \mathbf{A}_1)^{-1}$.

Let $\mathbf{b}'_1$ be the vector where $b'_{1i} = 1$ if $i$ is an outside user and $b'_{1i} = 1 - \beta$ if $i$ is an inside user. The value $(1 - \beta)E'_I + E'_O$ is exactly equal to

$$\langle \mathbf{c}'_1, \mathbf{b}'_1 \rangle = (\mathbf{c}'_1)^\top \mathbf{b}'_1 = (\beta\mathbf{1})^\top (\mathbf{I} - \mathbf{A}_1)^{-1} \mathbf{b}'_1.$$

Another interpretation of this is the engagement derived in the limiting state after injecting $(1 - \beta)$ of type 1 to inside users and 1 unit of type 1 to outside users. We first claim that $(I - \mathbf{A}'_1)^{-1}\mathbf{b}'_1 = \mathbf{1}$, the all ones vector. Indeed, such a solution is the unique $\mathbf{y}$ that satisfies $(I - \mathbf{A}'_1)\mathbf{y} = \mathbf{b}'_1$. Notice that for the all ones vectors, for outside users, the $i$'th row of $\mathbf{A}'_1$ is all 0s, so the term is 0, and for inside users, the $i$'th row has FOLLOWING($i$) number of terms each with value $\frac{\beta}{\text{FOLLOWING}(i)}$, so the corresponding component is $1 - \beta$. Therefore, $\mathbf{1}$ is the limiting state, and the engagement is exactly $(\beta\mathbf{1})^\top \mathbf{1} = n\beta$. This equality implies $(1 - \beta)E'_I + E'_O \le \beta n$, as needed.

From Inequality (3), by simple algebra, we get

$$n + E_I \ge \frac{E_I + E_O}{\beta} = \frac{OPT^{eng}}{\beta}.$$

Using this inequality, we have

$$nT\delta\alpha + (1 - T\delta)E_O + (1 - T\delta + T\delta\alpha)E_I$$
$$= nT\delta\alpha + T\delta\alpha E_I + (1 - T\delta)OPT^{eng}$$
$$= T\delta\alpha(n + E_I) + (1 - T\delta)OPT^{eng}$$
$$\ge \frac{T\delta\alpha OPT^{eng}}{\beta} + (1 - T\delta)OPT^{eng}$$
$$= \left(1 - T\delta + \frac{T\delta\alpha}{\beta}\right) OPT^{eng}$$
$$= \left(1 - T\delta \left(1 - \frac{\alpha}{\beta}\right)\right) OPT^{eng},$$

as needed.

To show tightness, consider an empty graph $G$ where, for all users $i$,

$$p_{ti} = \begin{cases} \beta & \text{if } t = 1 \\ \max\left\{\alpha - \frac{\beta - \alpha}{T-1}, 0\right\} & \text{if } t \neq 1. \end{cases}$$

For ease of notation, we let $\gamma = \max\left\{\alpha - \frac{\beta - \alpha}{T-1}, 0\right\}$. Notice that the maximum retweet probability is $\beta$ and $\alpha - \frac{\beta - \alpha}{T-1}$ is the exact value for other types to get the average retweet probability to $\alpha$,

$$\frac{\beta + (T-1)\left(\alpha - \frac{\beta - \alpha}{T-1}\right)}{T} = \frac{\beta + (T-1)\alpha - \beta + \alpha}{T} = \alpha,$$

so the true average is

$$\frac{\beta + (T-1)\gamma}{T} = \max\left\{\frac{\beta}{T}, \alpha\right\} \geq \alpha.$$

Further, notice that since the graph is disconnected, the limiting state is exactly the injection policy. In this case, one policy is to inject only type 1 which gives engagement $n\beta$. Hence, $OPT^{eng} \geq n\beta$. To be $\delta$-diverse, a policy must inject at least $\delta$ of each type to all users and can therefore inject at most $1 - (T-1)\delta$ to type 1. Hence,

$$OPT^\delta \leq \left(\underbrace{\beta(1 - (T-1)\delta)}_{\text{type 1}} + \underbrace{(T-1)\delta\gamma}_{\text{other types}}\right)n.$$

Plugging these inequalities in, we have

$$\begin{aligned} \text{cost}^\delta(G, \mathbf{p}) &= 1 - \frac{OPT^\delta}{OPT^{eng}} \\ &\geq 1 - \frac{(\beta(1 - (T-1)\delta) + (T-1)\delta\gamma)n.}{\beta n} \\ &= 1 - \left(1 - (T-1)\delta + \frac{(T-1)\delta\gamma}{\beta}\right) \\ &= (T-1)\delta\left(1 - \frac{\gamma}{\beta}\right) \\ &= T\delta\frac{T-1}{T}\left(1 - \frac{\gamma}{\beta}\right) \\ &= T\delta\left(1 - \frac{1}{T} - \frac{(T-1)\gamma}{T\beta}\right) \\ &= T\delta\left(1 - \frac{\beta + (T-1)\gamma}{T} \cdot \frac{1}{\beta}\right) \\ &= T\delta\left(1 - \max\left\{\alpha, \frac{\beta}{T}\right\}\frac{1}{\beta}\right) \\ &= T\delta\left(1 - \max\left\{\frac{\alpha}{\beta}, \frac{1}{T}\right\}\right) \\ &= \min\left\{T\delta\left(1 - \frac{\alpha}{\beta}\right), (T-1)\delta\right\}. \qquad \square \end{aligned}$$

One special case of particular interest is when users are homogeneous, that is, all having the same retweet probabilities, say $p_1 \geq \cdots \geq p_T$. In this case, as long as we were not in the degenerate case where $p_t = 0$ for all $t$, the bound simplifies to: $\text{cost}^\delta(G, \mathbf{p}) \leq \delta(\sum_{t \neq 1}(1 - \frac{p_t}{p_1}))$. This bound is at least as strong as the theoretical worst case and strictly stronger when it is not the case that $p_2 = \cdots = p_t = 0$.

We also note that, as the proof of Theorem 1 shows, the bound is tight in an empty graph (with no edges). It is also not hard to establish the upper bound in such a graph. The power of Theorem 1, then, lies in showing that the engagement-diversity tradeoff is no worse when generalizing an empty graph to an arbitrary social network with elaborate retweet dynamics.

## 4 THE ENGAGEMENT-DIVERSITY TRADEOFF IN PRACTICE

This section aims to use data to infer practical settings of $G$ and $\mathbf{p}$ and gain an empirical understanding of the cost of $\delta$-diversity.

*Data processing.* We reconstruct the social network ($G$) of users and their retweet probabilities ($\mathbf{p}$) with respect to different tweet types from Twitter datasets, which have been analyzed and evaluated in previous studies [10–12]. They consists of tweets posted within a week after certain political events. Here, we specifically focus on two. First, tweets relating to gun control the week after June 16, 2016, when there was a democratic filibuster for gun control reforms (we refer to this as the Gun Control dataset), and second, tweets relating to abortion the week after June 30, 2016, when the U.S. Supreme court struck down Texas restrictions (we refer to this as the Abortion dataset). We refer the curious reader to Garimella et al. [11] for in-depth details about the datasets.

Briefly, the datasets contain a collection of tweets along with information including the tweet text, the ID of the Twitter user that tweeted (or retweeted), and the hashtags mentioned. Additionally, the datasets include each user's social relations, i.e., which other users they follow. This allows us to directly reconstruct the follower graph $G$. In all, the Gun Control dataset has 3,975 users while the Abortion dataset has 7,284 users. The former has a total of 945,286 edges and the latter a total of 1,880,679 edges, meaning users followed roughly 238 and 258 others on average, respectively. The distributions of how many users each user follows and is followed by can be found in Figure 2 located in Appendix B.

We use hashtags to classify the tweets in the dataset into types, allowing us to later infer retweet probabilities. In particular, as the number of distinct hashtags is extremely large (e.g., more than 695,000 in the Abortion dataset), we restrict the analysis to the 2,000 most common hashtags. Using this, we construct a hashtag network, where a link between two hashtags indicates that these hashtags appeared in the same tweet and are, therefore, related [1, 22]. We then use this network to cluster hashtags into a more manageable number of "types." Specifically, we use the Louvain algorithm [6] on the hashtag network to extract its community structure, where each community of hashtags defines a specific type. This gives us a classification of hashtags into five and four distinct types in the two datasets.

We then use hashtag occurrences as a proxy for the number of tweets of a certain type. We first compute the number of times a user retweets a type by counting the times a corresponding hashtag appeared. Note that, generally, a single tweet may be considered part of multiple types or increase the count of a particular topic by more than one, as the tweet may contain multiple hashtags. To determine the number of tweets of a particular type *seen* by a user, we count the number of corresponding hashtag occurrences in their neighbors' tweets, both original and retweets.

Finally, we use two methods to infer retweet probabilities using the counts. First, we simply take the proportion of retweets divided by the number of tweets seen of a type. We call these "mode probabilities." For the second, we assume that the dataset is just a single observation of a user's retweet probability and use Bayesian updating. Specifically, we assume an independent Beta(1, 100) prior on users' retweet probabilities, which reasonably closely corresponds to the distribution of retweet probabilities observed. We then do Bayesian updating to get a posterior distribution on retweet probabilities; this has the convenient property that if a user retweets $r$ out of a total of $s$ seen, the posterior is Beta($1 + r$, $100 + s$).

We take two samples from the Beta distribution for each user. Together with the mode probabilities, we end up with three instances for each dataset that share the same graph but have distinct retweet probabilities.

*Experiments.* For each of the instances, we directly analyze the engagement-diversity tradeoff under the estimated retweet probabilities, as well as analyzing it under various modifications to these probabilities in order to understand the impact. First, we see the effect of scaling up the retweet probabilities. We directly multiply all inferred $A_{tij}$ probabilities by a constant, with factors 1, 3, 10, and 30 (1, of course, being the original inferred probabilities). After scaling, we always cap the maximum retweet probability to .99. The capping ensures both that the probabilities are consistent with their modeling definition and that the linear system converges. Next, we consider scaling the $A_{tij}$s distinctly for each $i$ (i.e., by a distinct value $C_i$ for each $i$) such that $\sum_{t,j} A_{tij}$ is a fixed constant, either 0.1, 0.3, or 0.9. This ensures that the expected number of retweets seen by each user at any timestep is exactly this value.

For each instance, we compute the necessary values to run the engagement-optimal and $\delta$-diverse linear programs, i.e., compute the type matrices and find their limiting value. From this, we can immediately solve the first LP to obtain $OPT^{eng}$. Next, we solve the LPs for 10 evenly spaced values of $\delta$, $\frac{1}{10T}, \frac{2}{10T}, \dots, \frac{1}{T}$ which gives us $OPT^{\delta}$ for these values. With these values, we can plot $\text{cost}^{\delta}(G, \mathbf{p}) = 1 - \frac{OPT^{\delta}}{OPT^{eng}}$. In the plots, we also include the theoretical worst-case bound of $(T - 1)\delta$.

Due to the large size of the dataset, these experiments are computationally intensive. All linear programs were solved using Gurobi on an Amazon Web Services (AWS) instance with 128 vCPUs of a 3rd Gen AMD EPYC running at 3.6GHz equipped with 1TB of RAM. Giving a Gurobi solver three threads, it takes on the order of 30 hours to compute the optimal values of the 10 LPs with $\delta = \frac{1}{10T}, \dots, \frac{1}{T}$. We ran all of our experiments in parallel, which used approximately 500 GB of RAM during the solve.

*Results.* Results are shown in Figure 1, with the second Beta sample for each instance, which is essentially identical to the first, relegated to Appendix B. The different colored lines correspond to the different scaling factors along with the theoretical lower bound derived in Section 3 of $(T - 1)\delta$.

As a general way to interpret the results, we consider the perspective of a policy-maker analyzing the ABORTION dataset. If they were willing to sacrifice 5% of engagement in order to boost diversity, with small probabilities, they may expect to be able to get $\approx 0.03$-diversity in the abortion dataset while for scaled up values

they may expect above 0.06-diversity (these numbers are $\approx 0.025$ and 0.045 for GUN CONTROL). We can understand these values of diversity as a proportion of the amount injected. A value of $\delta = 0.06$ means the proportion of tweets seen of each type is at least 6% of the magnitude of injected tweets. Although this may seem low, as there are four types, 24% of the magnitude is necessarily accounted for by simply showing diverse content. If the policy-maker is willing to sacrifice 10%, this number shoots up to about 0.07-diversity for lower scales and above 0.10 for higher, i.e., over 40% of the magnitude already accounted for (0.04, 0.07 and 35% for GUN CONTROL).

Finally, another interesting observation is that both increasing the scale and increasing the expected number of retweets very reliably improves the tradeoff.

## 5 DISCUSSION

While our model may appear stylized, we believe that it is quite robust. In essence, the main assumptions are a partition of tweets into types and known retweet probabilities, both of which seem quite reasonable.

Some issues that are ostensibly outside the scope of our model can, in fact, be captured by it. One is that not all tweets that fall into even a specific type, such as "climate change," have the same retweet probabilities. In theory, one could make the set of types arbitrarily granular, but this would make diversity constraints impractical. A better approach, which we believe to be plausible, is to set $p_{ti}$ (the probability of user $i$ retweeting type $t$) to be the *average* of the retweet probabilities of user $i$ for different tweets that are included in type $t$.

Another seemingly restrictive modeling choice that can easily be relaxed is the fact that a user's engagement is defined with respect to their retweet probability rather than a distinct "type engagement" parameter (or "type affinity"). Differentiating these features would allow the model to capture users that perhaps have high engagement yet rarely retweet. We could have instead introduced an additional parameter $e_{ti}$ for each user and type to be used in the definition of engagement, i.e., $\text{ENG}(\mathbf{x}) = \sum_t \langle \mathbf{e}_t, \mathbf{x}_t \rangle$. At a technical level, this hardly seems to affect the model nor the results; our choice to exclude it was solely for presentation, as we did not believe the gain in generality was worth the loss in comprehensibility in a paper already defining half the alphabet.

That said, we readily acknowledge that our model has limitations. To name one, we view propagation dynamics in the social network as resulting from retweets of content that is injected by the platform. But users also create content; for example, a political reporter will likely write new tweets about politics. This can be modeled as another injection policy that is outside of our control, but it is unclear what values one would choose for this policy.

Nevertheless, in our view, our model and analysis provide useful insights into the engagement-diversity tradeoff. As discussed in Section 1, however, the jury is still out on the diversity-polarization connection, and it is a topic of intensive inquiry. With a better (quantitative) understanding of this connection, our results could be directly leveraged to analyze engagement-polarization tradeoffs, potentially helping social media platforms curb negative societal impacts.

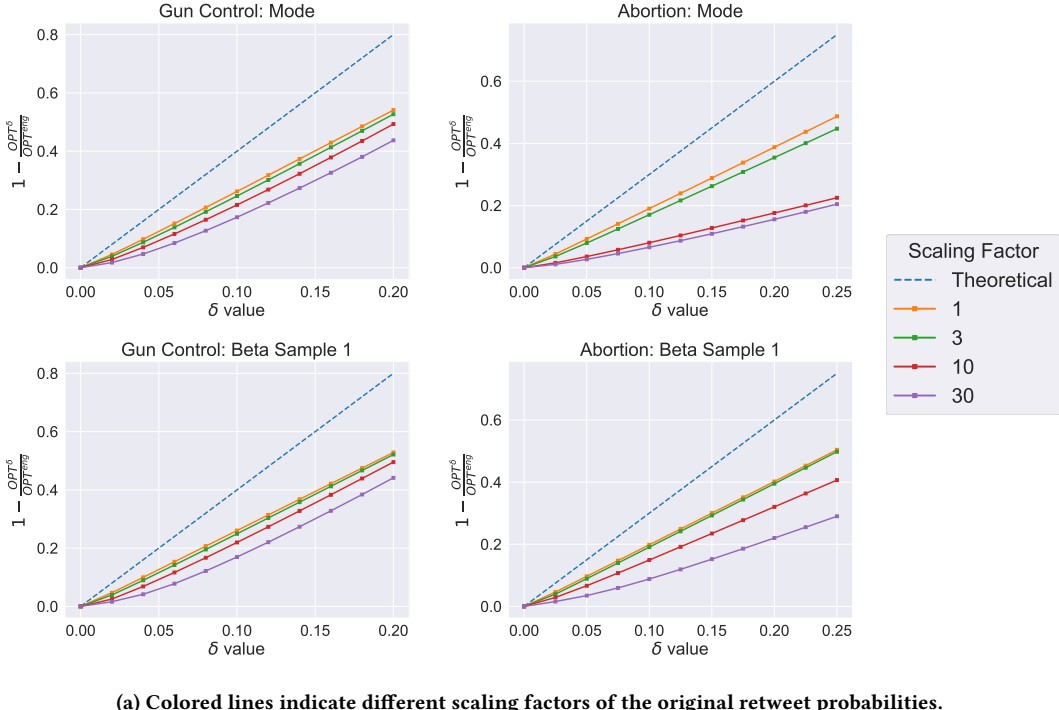

(a) Colored lines indicate different scaling factors of the original retweet probabilities.

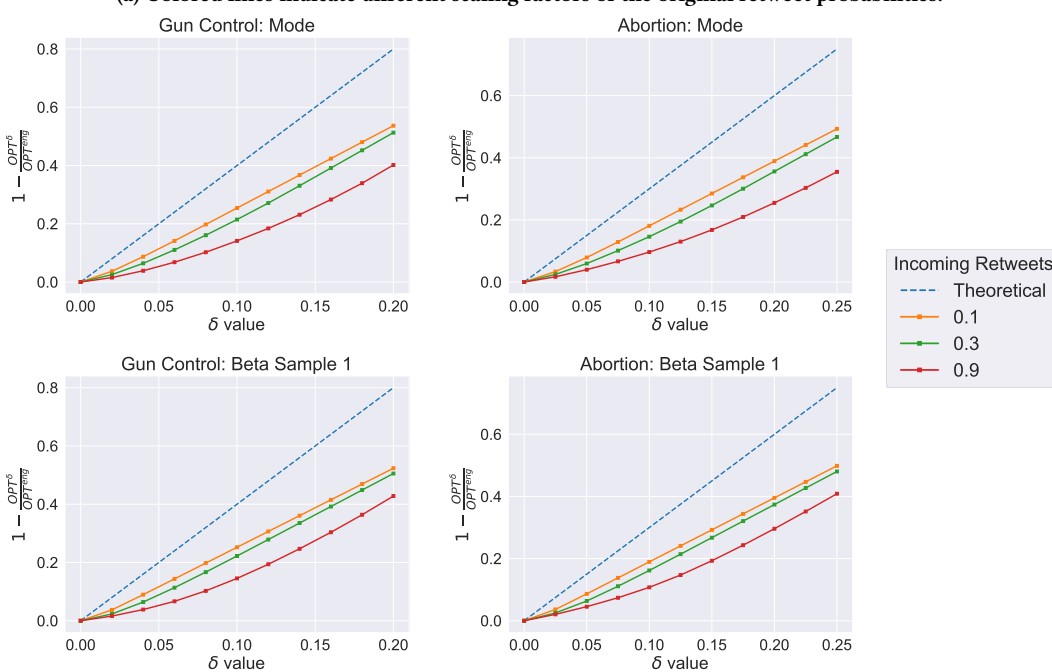

(b) Colored lines indicate the expected number of incoming retweets for each agent.

Figure 1: Plots of $1 - \frac{OPT^\delta}{OPT^{eng}}$ in various settings. The left are based on the GUN CONTROL dataset, while the right are based on the ABORTION one. Mode probabilities were inferred by directly calculating the ratio between retweeted and inferred. Beta samples were samples from a posterior distribution of retweet probabilities. The blue line in each diagram indicates the theoretical lower bound from Section 3; it is equal to $(T-1)\delta$ where $T = 4$ for the abortion dataset, and $T = 5$ for the gun control one. Values of $1 - \frac{OPT^\delta}{OPT^{eng}}$ are computed for values of $\frac{i}{10 \cdot T}$ for $i = 0, \ldots, 10$.

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

# Appendix

## A TIME-INDEPENDENT VS. TIME-DEPENDENT STRATEGIES

In Section 2, we mentioned that our model is essentially equivalent to one that has a time horizon $K$, engagagement is defined as the average engagement over timesteps, diversity is satsified at every timestep, and strategies can be time dependent. Here we formalize and prove this claim.

THEOREM 2. *Fix $\delta$ and let $\mathbf{b}^*$ be a solution to the $\delta$-diversity program. Fix a time horizon $K$ and let $\mathbf{x}^{(0)}, \ldots, \mathbf{x}^{(K)}$ be the states induced by injecting $\mathbf{b}^*$ at every timestep. Using the notation $ENG^{av}\left(\mathbf{x}^{(0)}, \ldots, \mathbf{x}^{(K)}\right) = \frac{1}{K+1} \sum_{k=0}^{K} ENG(\mathbf{x}^{(k)})$, we then have:*

*(1) The notions of engagement on $\mathbf{b}^*$ converge: $ENG^{av}\left(\mathbf{x}^{(0)}, \ldots, \mathbf{x}^{(K)}\right) = (1 - O(1/K)) \cdot ENG(\mathbf{b}^*)$,*

*(2) Diversity approaches $\delta$ exponentially fast: $DIV(\mathbf{x}^{(k)}) \geq \delta - \frac{1}{\exp(\Omega(k))}$,*

*(3) The policy $\mathbf{b}^*$ achieves approximately-optimal engagement. That is, if there is a strategy $\mathbf{b}^{(0)}, \ldots, \mathbf{b}^{(K)}$ inducing a sequence $\mathbf{y}^{(0)}, \ldots, \mathbf{y}^{(K)}$ such that $DIV(\mathbf{y}^{(k)}) \geq \delta$ for all $k$, then $ENG^{av}\left(\mathbf{y}^{(0)}, \ldots, \mathbf{y}^{(K)}\right) \leq (1 + O(1/K)) \cdot ENG^{av}\left(\mathbf{x}^{(0)}, \ldots, \mathbf{x}^{(K)}\right)$.*

The proof of Theorem 2 relies on the following simple lemma, a consequence of Gelfrand's formula [23], the main argument of which was proved by an anonymous user on Stackexchange,[6] although variations are clearly known in the literature. We nonetheless include the entire argument for completeness.

LEMMA 1. *There are constants $\lambda > 0$ and $\gamma \in (0, 1)$ depending only on $G$ and $\mathbf{p}$ such that for each type matrix $\mathbf{A}_t$, any injection policy $\mathbf{b}_t$, and any power $\ell \geq 0$, $\sum_{\ell=k}^{\infty} \|\mathbf{A}_t^\ell \mathbf{b}_t\|_1 \leq \lambda \gamma^k$.*

PROOF. Gelfand's formula implies that for any type $t$, $\lim_{\ell \to \infty} \|\mathbf{A}_t^\ell\|_1^{\frac{1}{\ell}} = \rho(\mathbf{A}_t)$, where $\rho(\mathbf{A}_t)$ is the spectral radius of $\mathbf{A}_t$ (see, e.g., Rudin [23], Theorem 10.35). As $\rho(\mathbf{A}_t) < 1$ for all types $t$, we can choose $\gamma \in (\max_t \rho(\mathbf{A}_t), 1)$ and doing so will imply $\lim_{\ell \to \infty} \|\mathbf{A}_t^\ell\|_1^{\frac{1}{\ell}} < \gamma$. means that for each type $t$, for sufficiently large $\ell$, it holds that $\|\mathbf{A}_t^\ell\|_1 < \gamma^\ell$. Hence, we can choose $M > 0$ large enough so that $\|\mathbf{A}_t^\ell\|_1 < M\gamma^\ell$ for all $\ell$ and $t$. Finally, observing that $\|\mathbf{b}_t\|_1 < n$ since it is a valid injection strategy, we have,

$$\sum_{\ell=k}^{\infty} \|\mathbf{A}_t^\ell \mathbf{b}_t\|_1 \leq \sum_{\ell=k}^{\infty} \|\mathbf{A}_t^\ell\|_1 \cdot \|\mathbf{b}_t\|_1$$

$$< n \sum_{\ell=k}^{\infty} M\gamma^\ell$$

$$= \left(Mn(1-\gamma)^{-1}\right)\gamma^k.$$

Choosing $\lambda = \left(Mn(1-\gamma)^{-1}\right)$ completes the proof. $\square$

We now prove the theorem.

PROOF OF THEOREM 2. Fix an instance $G = ([n], E)$, retweet probabilities $\mathbf{p}$, and a value $\delta \leq 1/T$. Fix a solution $\mathbf{b}^*$ to the $\delta$-diversity program, a time horizon $K$, and induced states $\mathbf{x}^{(0)}, \ldots, \mathbf{x}^{(K)}$. Fix the corresponding constants $\lambda$ and $\gamma$ from Lemma 1. Recall that $\mathbf{x}_t^{(k)} = \sum_{\ell=0}^{k} \mathbf{A}_t^\ell \mathbf{b}^*$.

We first consider part (1). Recall that $\mathbf{x}(\mathbf{b}^*) = \sum_{k=0}^{\infty} \mathbf{A}^\ell \mathbf{b}^*$ and hence $\mathbf{x}(\mathbf{b}^*)$ dominates $\mathbf{x}^{(k)}$ component-wise. Since $ENG(\mathbf{x})$ is monotonic in the components of $\mathbf{x}$, this means that $ENG(\mathbf{b}^*) \geq ENG(\mathbf{x}^{(k)})$ for all $k$, so,

$$\frac{1}{K+1} \sum_{k=0}^{K} ENG(\mathbf{x}^{(k)}) \leq \frac{1}{K+1} \sum_{k=0}^{K} ENG(\mathbf{b}^*) = ENG(\mathbf{b}^*).$$

In the degenerate case where $ENG(\mathbf{b}^*) = 0$, (1) immediately follows as both sides are equal to 0. Hence, we now consider the case where $ENG(\mathbf{b}^*) > 0$. Notice that $ENG(\mathbf{b}^*)$ does not depend on $K$ and is hence a constant in the $O()$ formula, so, rearranging the statement, it suffices to show that

$$ENG(\mathbf{b}^*) - ENG^{av}\left(\mathbf{x}^{(0)}, \ldots, \mathbf{x}^{(K)}\right) = O(1/K).$$

For the rest of the proof, it will be useful to observe that $\mathbf{x}^{(k)}$ converges to $\mathbf{x}(\mathbf{b}^*)$. More formally, using Lemma 1, we have that for all types $t$ and times $k$,

$$\left\|\mathbf{x}(\mathbf{b}^*)_t - \mathbf{x}_t^{(k)}\right\|_1 = \left\|\sum_{\ell=0}^{\infty} \mathbf{A}_t^\ell \mathbf{b}_t^* - \sum_{\ell=0}^{k} \mathbf{A}_t^\ell \mathbf{b}_t^*\right\|_1 = \left\|\sum_{\ell=k+1}^{\infty} \mathbf{A}_t^\ell \mathbf{b}_t^*\right\|_1 \leq \lambda \gamma^{k+1} \tag{4}$$

---

[6]https://math.stackexchange.com/questions/2561701/bound-on-the-norm-of-a-matrix-power

Additionally, we observe that $\langle \mathbf{p}_t, \mathbf{x}_t \rangle \leq \|\mathbf{x}_t\|_1$ for all $\mathbf{x}$ because each component of $\mathbf{p}_t < 1$. By combining these facts and expanding definitions, it follows that:

$$\text{ENG}(\mathbf{b}^*) - \text{ENG}^{av}\left(\mathbf{x}^{(0)}, \ldots, \mathbf{x}^{(K)}\right) = \sum_t \langle \mathbf{p}_t, \mathbf{x}(\mathbf{b}^*)_t \rangle - \frac{1}{K+1} \sum_{k=0}^{K} \sum_t \langle \mathbf{p}_t, \mathbf{x}_t^{(k)} \rangle$$

$$= \frac{1}{K+1} \sum_{k=0}^{K} \sum_t \langle \mathbf{p}_t, \mathbf{x}(\mathbf{b}^*)_t \rangle - \frac{1}{K+1} \sum_{k=0}^{K} \sum_t \langle \mathbf{p}_t, \mathbf{x}_t^{(k)} \rangle$$

$$= \frac{1}{K+1} \sum_{k=0}^{K} \sum_t \langle \mathbf{p}_t, \mathbf{x}(\mathbf{b}^*)_t - \mathbf{x}_t^{(k)} \rangle$$

$$\leq \frac{1}{K+1} \sum_{k=0}^{K} \sum_t \left\| \mathbf{x}(\mathbf{b}^*)_t - \mathbf{x}_t^{(k)} \right\|_1$$

$$\leq \frac{1}{K+1} \sum_{k=0}^{K} \sum_t \lambda \gamma^{k+1}$$

$$= \frac{t}{K+1} \sum_{k=0}^{K} \lambda \gamma^{k+1}$$

$$\leq \frac{t}{K+1} \sum_{k=0}^{\infty} \lambda \gamma^{k+1}$$

$$= \frac{t \lambda \gamma}{(1-\gamma)(K+1)} = O(1/K).$$

Next, we consider part (2), which follows more straightforwardly from Equation (4). Since $\text{DIV}(\mathbf{x}(b^*)) \geq \delta$ by assumption, each component of $\mathbf{x}(b^*) \geq t$. Since no component can differ by more than the $L_1$ distance between vectors, each component of $\mathbf{x}_t^{(k)}$ is at least $\delta - \lambda \gamma^{k+1}$ by Inequality (4). Since $\gamma < 1$, $\lambda \gamma^{k+1} = \frac{1}{\exp(\Omega(k))}$, as needed.

We now move on to part (3). Fix a $\delta$-diverse strategy $\mathbf{b}^{(0)}, \ldots, \mathbf{b}^{(K)}$. Consider $\mathbf{b}^{av} = \frac{1}{K+1} \sum_{k=0}^{K} \mathbf{b}^{(k)}$, the average of the time-dependent injections. First, observe that $\mathbf{b}^{av}$ is, in fact, a valid injection policy (i.e., nonnegative with no user shown more than one unit) since it is the linear combination of valid injection policies. We use $\mathbf{b}^{av}$ to more directly compare the time-dependent strategy to $\mathbf{b}^*$.

We begin by showing that $\mathbf{x}(\mathbf{b}^{ab})$ component-wise dominates $\frac{1}{K+1} \sum_{k=0}^{K} \mathbf{y}^{(k)}$, i.e., $x(\mathbf{b}^{ab})_{ti} \geq \frac{1}{K+1} \sum_{k=0}^{K} y_{ti}^{(k)}$ for all $t$ and $i$. To that end, we unravel the recursive definition of $\mathbf{y}_t^{(k)}$. We have

$$\mathbf{y}_t^{(k)} = \sum_{\ell=0}^{k} (\mathbf{A}_t)^{k-\ell} \mathbf{b}_t^{(\ell)}.$$

Plugging that into the linear combination,

$$\frac{1}{K+1} \sum_{k=0}^{K} \mathbf{y}_t^{(k)} = \frac{1}{K+1} \sum_{k=0}^{K} \sum_{\ell=0}^{k} (\mathbf{A}_t)^{k-\ell} \mathbf{b}_t^{(\ell)}.$$

Notice that a term $\mathbf{A}_t^a \mathbf{b}_t^{(b)}$ with a specific combination of $a$ and $b$ can only appear at most once, only when the outside sum has $k = a + b$. Hence, since all the summands are nonnegative, we have

$$\frac{1}{K+1} \sum_{k=0}^{K} \mathbf{y}_t^{(k)} \leq \frac{1}{K+1} \sum_{\ell=0}^{K} \sum_{k=0}^{K} (\mathbf{A}_t)^{\ell} \mathbf{b}_t^{(k)}$$

$$= \sum_{\ell=0}^{K} (\mathbf{A}_t)^{\ell} \left( \frac{1}{K+1} \sum_{k=0}^{K} \mathbf{b}_t^{(k)} \right)$$

$$= \sum_{\ell=0}^{K} (\mathbf{A}_t)^{\ell} \mathbf{b}_t^{av}$$

$$\leq \sum_{\ell=0}^{\infty} (\mathbf{A}_t)^{\ell} \mathbf{b}_t^{av}$$

$$= \mathbf{x}(\mathbf{b}^{av})_t$$

with $\leq$ defined component-wise. Using this, we have that

$$\text{ENG}(\mathbf{b}^{av}) \geq \frac{1}{K+1} \sum_{k=0}^{K} \text{ENG}(\mathbf{y}^{(k)}) = \text{ENG}^{av}\left(\mathbf{y}^{(0)}, \ldots, \mathbf{y}^{(K)}\right).$$

Further, notice that since for each $k$, $\text{DIV}(\mathbf{y}^{(k)}) \geq \delta$, $\text{DIV}\left(\frac{1}{K+1}\sum_{k=0}^{K} \mathbf{y}^{(k)}\right) \geq \delta$, so again by the component-wise domination, $\text{DIV}(\mathbf{x}(b^{av})) \geq \delta$. This implies that $\mathbf{b}^{av}$ is a feasible solution to the $\delta$-diversity program. Hence, by the optimality of $\mathbf{b}^*$, $\text{ENG}(\mathbf{b}^{av}) \leq \text{ENG}(\mathbf{b}^*)$. Putting it all together, we have

$$\text{ENG}^{av}\left(\mathbf{y}^{(0)}, \ldots, \mathbf{y}^{(K)}\right) \leq \text{ENG}(\mathbf{b}^{av}) \leq \text{ENG}(\mathbf{b}^*) \leq (1 + O(1/K))\text{ENG}^{av}\left(\mathbf{x}^{(0)}, \ldots, \mathbf{x}^{(K)}\right),$$

where the last inequality follows from part (1). □

# B  OMITTED FIGURES

Here we include some figures that were omitted from Section 4.

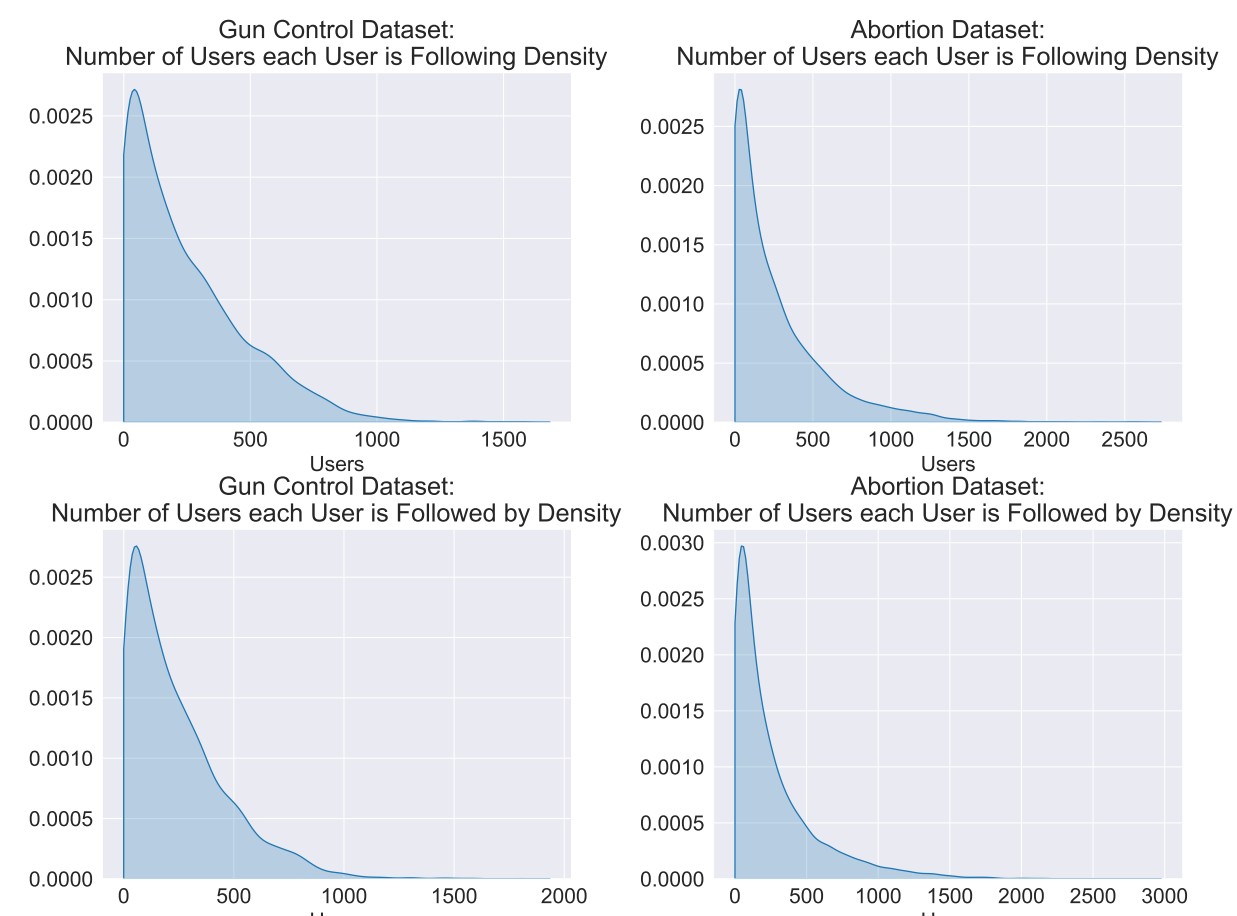

**Figure 2: Densities of the number of users each user follows and is followed by in each dataset. Densities were computed using kernel density estimation.**

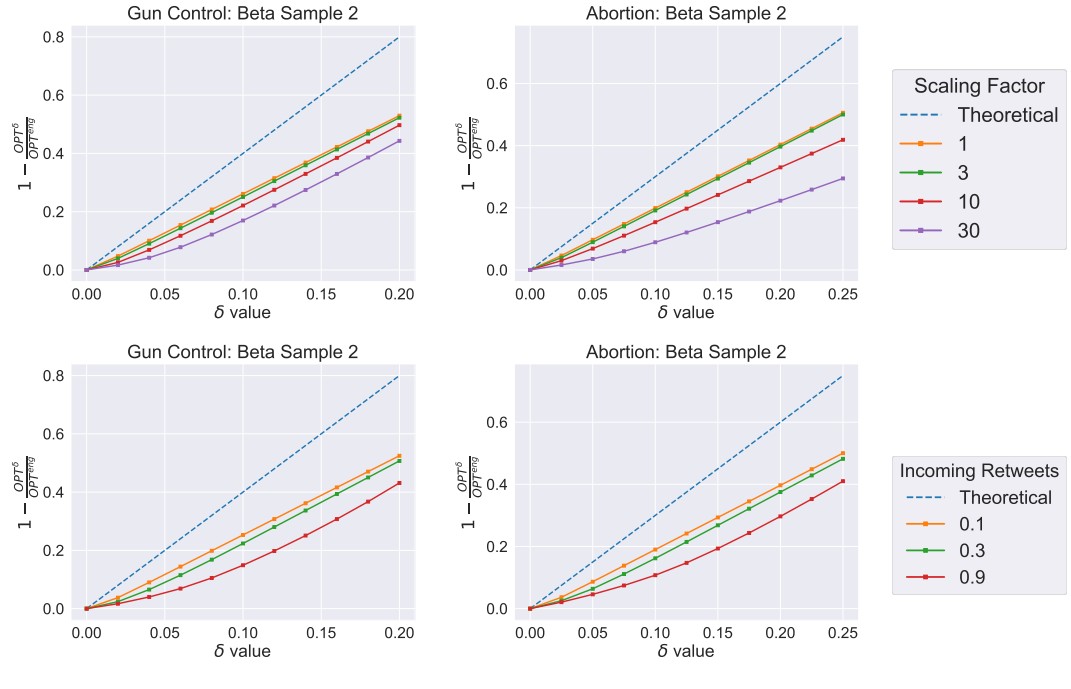

Figure 3: The second Beta sample from each setting that was omitted from Figure 1.

