# OpenReview forum: "Optimal Engagement-Diversity Tradeoffs in Social Media"
_ACM.org/TheWebConf/2024/Conference — TheWebConf24_

### Official Review · Reviewer_4Le7 · 2023-11-23

**Novelty:** 5
**Technical Quality:** 4

**Review:**

This paper studies optimal engagement-diveristy tradeoffs in social media, via a content injection policy that injects contents of diverse types to users, even at the cost of lower engagement. They model not just first-step engagement but also how content propagates on a network. They formulate a linear program to find an injection policy, and theoretically bound the tradeoff.

The work is interesting in formulating such a tradeoff with propagation of engagement to followers. The experiment is reasonable, though with some limitations as I ask about next:
 - what are the retweet dynamics induced by your parameter estimation? Do they compare with real-world dynamics? In other words, how accurate is your propagation model?

 - More generally, what intuitively do we gain by the propagation model? What changes in the optimal solution? How robust is the solution to errors in the propagation model or its estimation?

 - I do not understand the .06 or .03 -diversity numbers in the context of the model. How much "additional" diversity do we get by injecting the tweets, over not showing the content?

Edit after author response: I have read the author response. I now understand the diversity number, I think; it would be good to know what the baseline without the injection would be. However, I still do not understand fully what the effect of the propagation model is on the injection policy, or whether the results would change if the author did not model propagation, only the first step effect of showing a user an item. This may be a relevant question for the effectiveness of the method in practice.

**Questions:**

Please answer the questions above

**Reviewer Confidence:**

3: The reviewer is confident but not certain that the evaluation is correct

**Scope:**

4: The work is relevant to the Web and to the track, and is of broad interest to the community

---

### Official Review · Reviewer_S5Mc · 2023-11-24

**Novelty:** 5
**Technical Quality:** 4

**Review:**

This paper studies the engagement-diversity tradeoffs in social media. The authors use the number of retweets to quantify the user engagement and use the minimum number of tweets of type $t$ seen by user $i$, across all tweet types $t$ and users $i$. They formulate the engagement optimization problem in a form of Linear Program. The main theoretical results are the theoretical upper bounds for the engagement-diversity tradeoff, derived from introducing two injection policies. The experiments investigate how the settings of probability affects the cost of $\delta$-diversity.

## Quality
The modeling and analysis are basically correct. The authors derive the upper bounds on the cost of engagement to reach $\delta$-diversity. However, it is quesitonable whether such a model is a good one (see Cons and Questions). The stylized model the authors use makes the derived bounds rather trivial.

## Clarity
The manuscript is generally clear. But the writing style is not favorable. There is no subsection in this paper. I suggest the authors to make use of subsections to organize the paper.

## Originality
This paper seems original.

## Significance
This paper investigates the engagement-diversity tradeoff in social media, which is an important topic in social media with lots of field experiments. It is not mentioned whether this paper is the first attempt to quantify and model the engagement-diversity tradeoff problem. Technically speaking, they provide a model and derive theoretical bounds for the problem. But the model is stylized and may not be general enough (see Cons and Questions).

## Pros (Strengths)
1. This manuscript is complete and self-contained.
2. This paper makes an attempt to model the engagement-diversity tradeoff problem and gives theoretical bounds of the cost to reach a certain amount of diversity.
3. The authors present their work objectively without overclaiming. They also point out the limitations of their model.

## Cons (Weaknesses)
1. The metrics used to measure the engagement and diversity do not seem reasonable. Using only the number of retweets to measure the engagement may not be enough, as other behaviors like "comment" and "like" also indicate user engagement in the system. The metric to measure diversity, as mentioned by the authors, is pessimistic. So it is questionable whether it can reflect the overall diversity in the system.
2. The model are too stylized and may be oversimplified. For example, the initial state is assumed to be the injected tweets. It is questionable whether the authors' analysis still applicable to other situation where the initial state $\mathbf{x}^{(0)}$ is different.
3. More thorough experiments are needed. For example, the datasets used here (Gun Control and Abortion) have similar average degree. Experiments on networks with different structural features are preferred.

**Questions:**

1. The authors focus on $\delta \leq 1/T$. It is hard to see whether this setting is reasonable. What is the common $\delta$ value in most social media?
2. It seems like there is no comparison of different policies in the experiment. So the two policies are just introduced for the theoretical bounds. Am I correct?
3. As I do not see any related work trying to model this problem in a mathematical way (most mentioned related work is about field experiments), are you the first to make this attempt?

Minor:
1. Line 38, Page 1. "such as [...]" lacks the references.
2. Line 768, Page 7. "improves" should be "improve".

**Reviewer Confidence:**

3: The reviewer is confident but not certain that the evaluation is correct

**Scope:**

4: The work is relevant to the Web and to the track, and is of broad interest to the community

---

### Official Review · Reviewer_uDzw · 2023-11-24

**Novelty:** 4
**Technical Quality:** 6

**Review:**

The paper considers a social media running an algorithm for deciding which news to propose to its members. The algorithm is designed in order to maximize the engagement of the users. However, it may enforce polarization. For this reason, the paper focuses on algorithms maximizing the engagement subject to satisfy some diversity constraint, such as to show at least a minimum fraction of news of different types. In particular, investigate the loss on engagement due to the introduction of this diversity constraint. The paper shows that this loss is essentially limited in settings in which the social platform is well-established.

I find the question investigated in this paper very interesting. Still, I have some doubts about the real significance of results due to the strong limitations in the model:
- engagement is measured only in terms of re-posting / re-twitting, but engagement in a social network also consists in views, likes, comments, new posts
- the models only considers content created by the platform, but not content created by the members of the social network
- the models only assume that the probability of re-twitting a post depends on the content of the post, and it is independent from the content of other received posts. However, most research in influence maximization and opinion formation, have observed and introduced dynamics for which the action of a node depends also on the number of message of a certain type that have been received.

It must be acknowledged that the first two limitations have been acknowledged by the same authors. Hence, they acknowledge that their results are not final, but a building block towards a realistic and useful model. For this reason, I am still prone to give some chances to the current paper.

**Questions:**

None

**Reviewer Confidence:**

3: The reviewer is confident but not certain that the evaluation is correct

**Scope:**

3: The work is somewhat relevant to the Web and to the track, and is of narrow interest to a sub-community

---

### Official Review · Reviewer_jrSC · 2023-11-25

**Novelty:** 5
**Technical Quality:** 5

**Review:**

## Overview
This paper proposes a model that examines whether the intentional introduction of diversity by social media platforms, such as Twitter, leads to a reduction in user engagement. One of the key drivers behind this study is the assertion that decreased diversity can lead to the creation of 'echo bubbles', where users are only exposed to a narrow range of similar content. The primary platform modelled in this paper is Twitter. The authors define user engagement as total retweets and diversity as the average number of 'seen' tweets of a particular type by a user. Furthermore, the paper assumes that a user could be exposed to potential retweets via two mechanisms: organic and algorithmic. Importantly, the platform can drive diversity through algorithmic exposure. A significant theoretical output is a bound on the maximum achievable engagement when platforms impose diversity under budget constraints. The authors prove this theorem in a particular steady-state setting of the system but the authors also prove more general cases in the appendix. Thus the main result is quite robust. The empirical section of the paper involves estimating parameters using Twitter data, whilst demonstrating how the simulations' results can guide decision-makers on the platform's desired level of diversity injection.

## Pros
- The modeling is realistic and accommodates strategies adopted by companies like Twitter and Facebook in their operational systems, which lends credibility to the results presented.
- To the previous I also add that the authors are thorough in the explanation of their modeling choices throughout the paper.
- The authors' analysis in the appendix of the model’s extension to changing interventions and solvability for non-steady states is appreciated as it broadens the model's potential.
- The method for determining the subset of 'T' types of tweets is intriguing. It would have been nice to see a bit more detail in the results
- The inclusion of empirical simulations and the explanation of how a decision-maker might utilize the findings is commendable. However, it could be clearer in interpreting the numbers.

## Cons
- The model's view of engagement overlooks other interaction types, such as comments or organic content creation. These could be potential improvements.
- The introduction asserts an academic consensus on the trade-off between engagement and diversity in online platforms. However, it was unclear whether the literature used to support this claim is sufficient, necessitating further evidence for this assertion.
- The tight engagement diversity tradeoff bound seems dependent on very specific conditions of the follower graph and user retweet probabilities. The actual applicability or usefulness of this tight bound under such conditions is uncertain.
- Increased exposure to different tweets may not necessarily yield universally positive outcomes, such as reduced polarization. The authors acknowledge this limitation, but it could be interesting to provide a sample of the tweets categorized under different types in the empirical part of the paper to better evaluate the logic of these types.
- This is related to the previous point, but I believe a bit more discussion could go into discussing how to define types. One for example could argue that even within the same type, some tweets could be more toxic and inflammatory, which leads to more negative outcomes. With the current model, we would be required to differentiate this into additional types. It is noteworthy that the authors mention that increasing the number of types is counterproductive so this is an area of further clarification.

**Questions:**

1. How might other types of engagement, such as comments, be integrated into the model? Could such integration potentially alter the results qualitatively?
2. Is the model robust against changes in the follower network ($A$ in the model) through time, which could significantly affect retweet dynamics? Is this scenario important?
3. The paper mentions that excessively increasing the number of possible tweet types could be counterproductive. How might an ideal number of types be determined, and what should these types represent?
4. Have alternative mechanisms for grouping tweets into types been tested for robustness?
5. Can the types be heavily correlated? Suppose one wants to separate left and right leaning tweets but also separate toxic tweets within those as a separate type.

**Ethics Review Description:**

No issues

**Reviewer Confidence:**

2: The reviewer is willing to defend the evaluation, but it is likely that the reviewer did not understand parts of the paper

**Scope:**

4: The work is relevant to the Web and to the track, and is of broad interest to the community

---

### Official Review · Reviewer_KsX4 · 2023-11-26

**Novelty:** 5
**Technical Quality:** 6

**Review:**

This paper presents an analysis of the trade-off between users’ engagement in social media and the diversity of information they see. The authors present a mathematical model of those factors, a theoretical analysis of their interplay, and some emprirical results.

The paper addresses an important problem with longstanding interest from science and societies. The theoretical model seems plausible, given especially the justification and discussion of modeling choices. The analysis seems sound and neat. Besides, I appreciate the authors’ effort in making the results understandable with empirical results explained in simple terms.

My questions are the following:
- Are there prior work on theoretical analysis of user engagement and/or information diversity? Given the importance of the topic, it's a little hard for me to believe that such work is completely missing in the literature.
- How do the four types of tweets look like in the two datasets?
- And how would the result change if the number of content types increases? This last question I believe is relevant to gain deeper insights with finer-granularity topics.

**Questions:**

See my questions above.

**Reviewer Confidence:**

2: The reviewer is willing to defend the evaluation, but it is likely that the reviewer did not understand parts of the paper

**Scope:**

4: The work is relevant to the Web and to the track, and is of broad interest to the community

---

### Decision · Program_Chairs · 2024-01-22

**Decision:**

Accept

**Comment:**

This paper presents an analysis of the trade-off between users’ engagement in social media and the diversity of information they see. The authors present a mathematical model of those factors, a theoretical analysis of their interplay, and some emprirical results. The reviewer agreed that it was an interesting question. They agreed that the model was reasonable, though several wished for different extensions, adjustments, or different baselines. Overall, this seems like a worthy contribution to the WEB conference.